# Mutation Hotspots Found in Bladder Cancer Aid Prediction of Carcinogenic Risk in Normal Urothelium

**DOI:** 10.3390/ijms24097852

**Published:** 2023-04-25

**Authors:** Sydney R. Grant, Li Tang, Lei Wei, Barbara A. Foster, Gyorgy Paragh, Wendy J. Huss

**Affiliations:** 1Department of Dermatology, Roswell Park Comprehensive Cancer Center, Buffalo, NY 14263, USA; 2Department of Cell Stress Biology, Roswell Park Comprehensive Cancer Center, Buffalo, NY 14263, USA; 3Department of Cancer Prevention & Control, Roswell Park Comprehensive Cancer Center, Buffalo, NY 14263, USA; 4Department of Biostatistics & Bioinformatics, Roswell Park Comprehensive Cancer Center, Buffalo, NY 14263, USA; 5Department of Pharmacology & Therapeutics, Roswell Park Comprehensive Cancer Center, Buffalo, NY 14263, USA

**Keywords:** urothelium, mutation hotspot, clonal mutations, carcinogenesis, bladder cancer, machine learning

## Abstract

More than 80,000 new cases of bladder cancer are estimated to be diagnosed in 2023. However, the 5-year survival rate for bladder cancer has not changed in decades, highlighting the need for prevention. Numerous cancer-causing mutations are present in the urothelium long before signs of cancer arise. Mutation hotspots in cancer-driving genes were identified in non-muscle-invasive bladder cancer (NMIBC) and muscle-invasive bladder cancer (MIBC) tumor samples. Mutation burden within the hotspot regions was measured in normal urothelium with a low and high risk of cancer. A significant correlation was found between the mutation burden in normal urothelium and bladder cancer tissue within the hotspot regions. A combination of measured hotspot burden and personal risk factors was used to fit machine learning classification models. The efficacy of each model to differentiate between adjacent benign urothelium from bladder cancer patients and normal urothelium from healthy donors was measured. A random forest model using a combination of personal risk factors and mutations within MIBC hotspots yielded the highest AUC of 0.9286 for the prediction of high- vs. low-risk normal urothelium. Currently, there are no effective biomarkers to assess subclinical field disease and early carcinogenic progression in the bladder. Our findings demonstrate novel differences in mutation hotspots in NMIBC and MIBC and provide the first evidence for mutation hotspots to aid in the assessment of cancer risk in the normal urothelium. Early risk assessment and identification of patients at high risk of bladder cancer before the clinical presentation of the disease can pave the way for targeted personalized preventative therapy.

## 1. Introduction

In 2023, there will be an estimated 82,290 new cases of bladder cancer, with an estimated 16,0 deaths, in the United States [1]. The most common clinical presentation of bladder cancer at the time of diagnosis is visible hematuria [2]. At diagnosis, cancers are classified as non-muscle-invasive bladder cancer (NMIBC) or muscle-invasive bladder cancer (MIBC). Most bladder cancers are NMIBC and are treated by resection with cystoscopy, with or without intravesical immuno- or chemotherapy. The current treatment regimens often cause a considerable physical, psychological, and economic burden. The costs for 5-year NMIBC surveillance range from $50,000 for low-risk patients to over $350,000 for high-risk patients [3]. More advanced stages of bladder cancer, such as MIBC, often require more aggressive treatments, including cystectomy combined with radiation or chemotherapy [4,5]. Despite advances in oncology, the 5-year survival rate for bladder cancer has not changed in the last 20 years [1], highlighting the need for new strategies in bladder cancer prevention.

Exposure to carcinogens is the highest risk factor for bladder cancer. The most common carcinogen exposures are tobacco and lifetime exposure to chemical byproducts [2,6]. The incidence of bladder cancer also increases with age [2,7]. Carcinogenesis is initiated long before the first clinical signs of cancer arise and requires the accumulation of multiple cancer-driving mutations [8]. Thus, similar to other carcinogen-exposed organs, clinically normal-appearing urothelium harbors numerous genomic mutations [9,10]. Several studies have found a positive selection of mutations in various genes, including *MLL2*, *KDM6A*, *ARID1A*, and *RBM10* [9,10].

Many studies have highlighted the mutational vulnerability of specific genomic regions within cancer-driving genes [11,12,13]. The regions more sensitive to mutations are referred to as mutation hotspots. Numerous hotspots related to bladder cancer diagnosis and prognosis have been identified, including hotspots in *TERT* [14,15,16], *TP53* [17,18], *PIK3CA* [15,17,18], and *FGFR3* [14,15,18]. A study comparing mutation hotspots among various stages of bladder cancer revealed an association of *TP53* mutations in more aggressive disease subtypes [15]. The presence of mutations at multiple hotspots has also been validated in normal urothelium [16,17], again supporting the occurrence of mutations before the manifestation of bladder cancer. Our previous publication shows mutation enrichment in normal epidermis within mutation hotspots of cutaneous squamous cell carcinoma [19]. However, no studies thus far provide a comprehensive comparison between highly mutated genomic regions of bladder cancer and normal urothelium. Although most cancer driver mutations cluster at hotspots, studies showing the utility of bladder cancer hotspot mutation analysis to predict cancer risk are lacking. Our work shows how mutation burden can help predict carcinogenic risk in clinically normal urothelium with no noticeable signs of disease. These findings provide data for the future development of genomic assays for individualized bladder cancer risk assessment and targeted bladder cancer prevention.

## 2. Results

### 2.1. Overlap between Highly Mutated Genomic Regions in MIBC and NMIBC

We have developed a software tool, hotSPOT, that locates genomic regions with high mutation frequency [20]. Somatic mutation datasets for IMPACT targeted sequencing of 103 NMIBC samples (MSK Eur Urol 2017) [21], and whole-exome sequencing of 409 MIBC samples (TCGA, 2017) [22] were run through hotSPOT as discovery cohorts to identify highly mutated genomic regions in both cancer sub-types. For unbiased comparison between the NMIBC and MIBC datasets, mutations considered from the all datasets were limited to the exome regions included in the MSK-IMPACT sequencing panel, which includes 5080 exons from 341 genes, covering a total of 922,349 bp [23]. HotSPOT was used to locate and rank genomic segments of 100 bp size containing one or more mutations. For each dataset, 10% of all segments containing the most mutations were ranked. Panels of the top 10% most mutated regions in NMIBC consisted of 105 segments (NMIBC10), and in MIBC consisted of 369 segments (MIBC10). The difference in segment quantity between NMIBC10 and MIBC10 is likely due to the increased number of available MIBC samples. There was no statistical difference in mutation frequency between the NMIBC and MIBC samples (*p* = 0.105, Wilcoxon Rank Sum Test). Several genomic areas contained large clusters of mutations, and, therefore, were covered by multiple genomic segments. A total of 44 genomic regions overlapped between the NMIBC10 and MIBC10 panels. There were 61 genomic regions unique to NMIBC10 and 322 regions unique to MIBC10. A panel of combined genomic regions from NMIBC10 and MIBC10 was also generated to represent the top 10% mutated regions in both types of bladder cancer (BC10). The obtained BC10 panel contained 427 unique genomic regions (Figure 1, Appendix A).

NMIBC10 and MIBC10 panels cover large genomic regions (10,500 bp and 36,900 bp, respectively), making it necessary to develop a refined list of genomic segments representing the most mutated regions of NMIBC and MIBC. Mutation frequency per sample was measured in each genomic segment separately for NMIBC and MIBC. One-way ANOVA was performed for each segment to compare the mutation count in bladder cancer samples amongst all other 100 bp regions within the ranked list. A Tukey HSD test was used to identify all regions that captured significantly more mutations than the majority of lower-ranked regions. This resulted in a final hotspot panel of 4 hotspots in NMIBC and 13 hotspots in MIBC (Figure 1). The 4 hotspots found in the NMIBC dataset were also present in the MIBC hotspot panel.

### 2.2. Correlation of Mutated Genomic Regions in Bladder Cancer and Normal Urothelium

To better understand the similarities between highly mutated regions in NMIBC and MIBC, we calculated the number of mutations captured in MIBC10 regions in both the NMIBC and MIBC datasets. We found a significant correlation (*p* < 0.0001), indicating that the mutation burden of individual genomic regions as measured with the MIBC10 panel was similar in both datasets. The same analysis using NMIBC10 regions also found a significant correlation (*p* < 0.0001) between the mutation numbers of the hotspot areas in the two different bladder mutations datasets (Figure 2A). These findings demonstrate that mutations are enriched in many common genomic regions in both NMIBC and MIBC.

We also assessed whether genomic regions frequently mutated in bladder cancer are also frequently mutated in normal urothelium and how the mutation burden in normal urothelium is associated with an individual’s risk of cancer. Previous studies have identified genomic hotspots in bladder cancer [15,17,18] and normal urothelium [14,16]. However, a robust comparison between the genomic hotspots in bladder cancer and normal urothelium is lacking. Furthermore, it is not known if hotspots can be used to identify individuals who are at a high or low risk of developing bladder cancer based on mutation burden in the clinically normal-appearing urothelium. To understand the similarities of highly mutated genomic regions in bladder cancer and clinically normal urothelium, mutation frequency was compared between normal urothelium samples from both healthy donors and bladder cancer patients vs. NMIBC and MIBC samples using BC10. Based on this analysis, a significant correlation (*p* < 0.0001) was found between the mutation frequency in BC10 genomic regions from normal urothelium and bladder cancer samples (Figure 2B).

A previous study that compared normal urothelium from donors with a history of bladder cancer (high risk of bladder cancer) showed an increase in mutations per exome compared with those with no history of bladder cancer (low risk of bladder cancer) [9]. Therefore, we hypothesized that highly mutated regions in high-risk samples showed a higher correlation with mutated regions in bladder cancer. Two publicly available whole-exome sequencing datasets of normal urothelium were identified to compare the mutation rate of hotspots in samples with low- and high-risk bladder cancer [9,10]. The low-risk normal urothelium cohort was comprised of 483 samples from post-mortem donors with no history of cancer from Lawson et al. [9]. The high-risk normal urothelium cohort included normal urothelium collected from individuals with bladder cancer during the time of cystectomy or nephroureterectomy, including 72 samples from Lawson et al. [9] and 161 samples from Li et al. [10].

To compare the mutation landscape between bladder cancer and high- or low-risk normal urothelium, the correlations were calculated between MIBC vs. low-risk normal urothelium and MIBC vs. high-risk normal urothelium within the MIBC10. A significant correlation was found for both comparisons; however, the correlation was more robust in the comparison between high-risk normal urothelium and MIBC samples (R = 0.425 vs. R = 0.140) (Figure 2C). In parallel, we calculated the correlation between NMIBC vs. low-risk normal urothelium and NMIBC vs. high-risk normal urothelium within the NMIBC mutated regions. However, a significant correlation was only found between NMIBC and high-risk normal urothelium (*p* < 0.001) (Figure 2D).

To determine whether the highly mutated regions in high- and low-risk urothelium are similar, the correlation of mutation burden between high- and low-risk normal urothelium within the NMIBC10 and MIBC10 genomic regions was calculated. No significant correlation was found within either of the genomic panels. Based on both panels, hotspots in *KDM6A* and *CDKN1A* were found to be higher in the low-risk normal urothelium, while multiple hotspots of *TP53* were found to be higher in the high-risk normal urothelium (Figure 2E). The apparent differences in mutation distribution of NMIBC10 and MIBC10 regions between high- and low-risk normal urothelium highlight the potential for these genomic regions as markers to differentiate high- and low-risk samples.

### 2.3. Mutation Hotspots in MIBC and NMIBC

Hotspots included in each hotspot panel were ranked based on the average number of mutations per 100 bp hotspot and compared against the remaining ranked genomic segments in NMIBC10 and MIBC10 (Figure 3A). A ranked list of genes was created based on each hotspot frequency to identify which genes were highly represented in the NMIBC and MIBC hotspot panels. The genes with hotspots in NMIBC were *ERBB2* (1), *FGFR3* (1), *PIK3CA* (1), and *TP53* (1). The genes with the most hotspots in MIBC were *TP53* (5), *FGFR3* (2), *CDKN1A* (1), *CDKN2A* (1), *ERBB2* (1), *KDM6A* (1), *NFE2L2* (1), and *PIK3CA* (Figure 3B). To further validate the relevance of the MIBC hotspot panel, an independent whole-exome sequencing dataset from 126 MIBC samples [10] was used to measure the mutation burden per sample within the hotspot panel and within an equivalent-sized panel of infrequently mutated genomic segments. Based on this analysis, there were significantly more mutations (*p* < 0.001) per sample within the MIBC hotspot panel (Figure 3C). These findings validate using the identified genomic segments as bladder cancer mutation hotspots. The mutation frequency per NMIBC and MIBC sample were measured for each hotspot in both panels. One hotspot in *TP53* was found to have a 3.40-fold increase in average mutation burden in MIBC compared with NMIBC samples (*p* < 0.05) (Figure 3D). However, two hotspots shared between the NMIBC and MIBC hotspot panels were found to have a significantly higher mutation burden in NMIBC samples compared with MIBC. One hotspot in *FGFR3* showed a 4.52-fold increase in average mutations per sample (*p* < 0.001) in NMIBC (Figure 3E), while another in *PIK3CA* showed a 1.52-fold increase in average mutations per sample (*p* < 0.05) in NMIBC (Figure 3F). However, as artificial differences in mutation burden between NMIBC and MIBC samples may be observed due to differences in methods of next-generation sequencing experiments, we tested the validity of the observed mutation differences in the three hotspots. Mutation frequency for each hotspot was measured in samples from the NMIBC samples and an independent dataset of 126 MIBC samples [10]. In the *TP53* hotspot, a 2.03-fold increase in average mutation burden in NMIBC vs. MIBC samples was observed, although this difference did not reach the level of statistical significance in the validation dataset (*p* = 0.21). However, a 4.21-fold increase in average mutations per sample was observed in NMIBC vs. NMIBC samples was shown in the *FGFR3* hotspot (*p* < 0.001) and a 2.36-fold increase in average mutations per sample in NMIBC vs. MIBC samples was shown in the *PIK3CA* hotspot (*p* < 0.05) (Appendix A).

### 2.4. Machine Learning Approach to Predicting Bladder Cancer Risk in Normal Urothelium Using Mutation Hotspots

To facilitate using large-scale genomic studies, a machine learning approach to model carcinogenic risk in normal urothelium was used. For the training and testing of our model, the two published datasets of high- and low-risk normal urothelium were used [9,10]. The combined dataset was randomly split 70%/30% into training and test datasets. Each sample in the training dataset was labeled as high or low risk and the mutation burden using the MIBC hotspot panel and NMIBC hotspot panel was calculated. Several personal risk factors for bladder cancer, including age, sex, and smoking status, were also considered in the models. Many published applications of machine learning models have shown that the use of personal risk factors in combination with biological data can improve model performance. Therefore, the personal characteristics known to affect the risk of bladder cancer were included, in addition to mutation hotspot burden, as predictive features for our model. To test the effectiveness of both the personal risk and hotspot features, three different combinations of features were tested: MIBC Hotspot + Personal Risk Factors, NMIBC Hotspots + Personal Risk Factors, and Personal Risk Factors Only. To compare the ability of each feature combination to accurately predict the cancer risk type, each training dataset was fitted to three different types of machine learning classification models. Logistic regression, neural networks, and random forest are the most commonly used machine learning models for assessing genomics and next-generation sequencing data [24,25]. To increase the robustness of our study, we utilized all three model types and compared model performance. The test dataset for each combination of features was then used to assess the model’s ability to classify normal urothelium samples based on carcinogenic risk correctly. Personal risk factors alone achieved the lowest AUC values for all three models, with 0.8786, 0.8429, and 0.8964 for logistic regression, neural network, and random forest, respectively. The combination of mutations in NMIBC hotspots and personal risk factors achieved AUC values of 0.8786, 0.8643, and 0.9000. The combination of mutations in MIBC hotspots and personal risk factors showed the best performance for all three models, yielding AUC values of 0.8929, 0.8643, and 0.9286. For all three combinations of features tested, the random forest model showed the strongest performance (Figure 4A). To compare the impact of each feature, the variable importance was calculated for each random forest model. All combinations of features showed age to have the highest importance, while the status of whether a patient was either a current or non-smoker was shown to have the lowest importance (Figure 4B). These findings suggest that the NMIBC and MIBC hotspot panels are able to aid the prediction of cancer risk of clinically normal urothelium.

## 3. Discussion

Cancer prevention is moving to the forefront of scientific and clinical interest in the field of oncology. Genomic analysis has the potential to shift our current focus on cancer diagnosis and treatment to a focus on cancer prevention. It is well known that cancer begins to develop long before the first clinical signs of disease [8,26,27]. Despite this knowledge, the early phases of carcinogenesis are understudied. Early identification of individuals at high risk of developing cancer would allow time to implement preventative interventions. Effective interventions may prevent individuals from ever developing cancer and reduce the treatment burden on those individuals. In this study, mutation hotspots were identified in bladder cancers as valuable tools for assessing carcinogenic risk in the clinically normal-appearing urothelium.

The use of next-generation sequencing has increased in recent years and has become a useful method for studying cancer genomics. However, high-depth sequencing is required to study mutations in clinically normal tissues, which is not currently economically feasible through whole-exome or whole-genome sequencing. Therefore, we have created the hotSPOT tool to identify optimal genomic targets for studying mutations in normal tissues. These studies demonstrate the ability of the hotSPOT tool [20] to identify highly mutated genomic regions among bladder cancer samples. All identified hotspots in NMIBC were also found to be present in MIBC, demonstrating the accumulation of mutation burden during cancer progression.

*TP53* was the gene with the largest number of hotspots in MIBC, and a hotspot in *TP53* was also present in the NMIBC hotspot panel. This observation is consistent with previous findings that *TP53* is the most frequently mutated gene with the highest number of mutations in MIBC. *TP53* loss-of-function mutations cause genomic instability and serve as drivers of MIBC [15,28]. While *TP53* loss-of-function mutations are known drivers of aggressive bladder cancer [29], mutations in *TP53* have not previously been associated with early disease or identified as risk factors in the normal urothelium. Somatic mutations in tumor suppressors can be found in all epithelia and can lead to clonal expansion; however, the significance of identifying clonal expansion-associated mutations is still controversial [30]. By middle age, clonal mutations represent half of the normal tissue in the esophagus [31]. In our work, we found not only a higher mutation burden in *TP53* hotspots in MIBC than in NMIBC, but that multiple hotspots of *TP53* were higher in the high-risk normal urothelium. These findings indicate that *TP53* mutations not only play a role in advanced bladder cancer but may serve as a marker of early carcinogenesis.

Hotspots in *CDKN1A*, *CDKN2A*, *KDM6A*, *NFE2L2* were unique to the MIBC hotspot panel. While mutations in these genes are associated with MIBC, recent studies show mutations in normal urothelium and NMIBC. Specifically, *CDKN1A* and *KDM6A* mutations are associated with MIBC [22]; however, recent analysis with a large number of NMIBC samples demonstrates frequent mutations in *CDKN1A* and *KDM6A* in NMIBC [32]. Mutations in *CDKN2A* are more common in MIBC [33], but have recently been identified in normal urothelium and tumor specimens from NMIBC patients [34] and deletion was associated with progression in NMIBC [35]. Mutations in *NFE2L2* (*Nrf2*) are not frequently described as common in bladder cancer; however, analysis of TCGA data of activating mutations found in the *Nrf2*/pathway are common in carcinogen-exposed cancers (including bladder) and are associated with aggressive disease [36]. Therefore, more analysis is needed to identify potential mutations associated with malignancy before tumor initiation.

In addition to *TP53*, hotspot mutations for *ERBB2*, *FGFR3*, and *PIK3CA* were in the NMIBC and MIBC panels. *ERBB2* mutations are less common in NMIBC than MIBC [33]. *FGFR3* mutations are detected in 60–70% of NMIBC cases [37], and *FGFR3* is known for mutation hotspots and translocations [38]. Mutations in *FGFR3* are associated with better prognosis and are considered highly targetable [38,39]; thus, identifying these mutations early could lead to prevention strategies. *PIK3CA* has mutated in approximately 20–30% of NMIBC and MIBC cases. *PIK3CA* mutations are associated with non-smokers, *FGFR3* mutations, and better outcomes in NMIBC [33,40]. Although all NMIBC hotspots were also found in MIBC, NMIBC mutation frequency in hotspots in *FGFR3* and *PIK3CA* was found to be significantly higher than in MIBC. This observation is potentially due to the change in coverage of clonal mutations throughout disease progression. While genes such as *FGFR3* and *PIK3CA* may have a high frequency in the early stages of bladder cancer, and are still detectable, they are outcompeted by cells with driver mutations associated with MIBC.

When assessing the correlation between high- and low-risk normal urothelium in the NMIBC10 and MIBC10, multiple hotspots in both *KDM6A* and *CDKN1A* genes showed higher mutation frequency in low-risk normal urothelium compared with high-risk normal urothelium. The high frequency of hotspot mutations in *KDM6A* and *CDKN1A* in low-risk normal urothelium further indicates that clonal groups of cells with specific mutations can provide initial growth advantage, but can be outcompeted by new, more aggressive clones during disease progression [41]. A further collection of normal urothelium mutation datasets with detailed follow-up of future cancer diagnosis will be necessary to validate the findings in this study and discover additional potential genomic markers of cancer risk.

DNA adducts, particularly 4-aminobiphenyl (cigarette smoke carcinogen) DNA adducts, are higher in the bladder cancers of smokers than in those of non-smokers [42]. A unique aspect of smoking is the introduction of CC > AA dinucleotide to certain cancer types, such as cancers of the lung and esophagus, although the induced mutations are less obvious in bladder cancers because DNA mutation analysis has been performed at low coverage [32]. Neither Lawson et al. nor Li et al. demonstrated a mutation signature associated with smoking [9,10]. Smoking status was not a robust predictive marker of either MIBC or NMIBC. However, this analysis needs further investigation as the Li et al. high-risk dataset did not include any current or former smokers within female patients [10]. However, the common use of aristolochic acid in Chinese patients resulted in a specific mutation pattern of T > A transversions in the 5′-CpTpG-3′ context [10].

Although our findings are compelling, some technical differences between the datasets must be acknowledged. MIBC samples required an average of 60% tumor cell nuclei for samples to pass quality standards, while NMIBC samples required only 40% tumor purity on histologic review. DNA was extracted from fresh frozen MIBC samples and from FFPE in NMIBC samples. Both studies utilized the Illumina HiSeq platform for DNA sequencing. Whole-exome sequencing with 2 × 76 bp paired-end reads was used for MIBC samples, while NMIBC samples were sequenced using the MSK-IMPACT targeted sequencing panel, which includes 2 × 100 bp paired-end reads [23]. Mutation calling was performed using the Firehose and MSK-IMPACT bioinformatics pipelines for MIBC and NMIBC, respectively [21,22]. Due to the variability of next-generation sequencing parameters and samples between the NMIBC and MIBC datasets, further analysis will be needed to validate this study’s findings on technically more uniform datasets.

In addition to the technical differences in the analysis of the datasets, the low- and high-risk populations analyzed were different. In Lawson et al., the population was European, and race was not documented [9], compared with the analysis of Chinese patients in Li et al. [10]. As race was not included in the Lawson et al. dataset, we did not include it in our analysis. Future studies could examine how *GSTM1* and *GSTT1* polymorphisms, which alter detoxification of certain carcinogens, alter the mutated genomic regions. *GSTM1*-null mutations affect up to 53% of all racial/ethnic groups, while *GSTT1*-null mutations affect up to 21% of Caucasian, 64% of Asian populations, and 45% of African populations [43,44].

Genomic mutations are known to arise in the earliest phases of carcinogenesis [27,30]. Populations of cells that harbor these mutations are scattered throughout tissues with no clinical signs of disease [8,19,26,45]. The concept of field carcinogenesis is supported by decades of clinical evidence [46]. Despite the long recognition of the mutation burden in normal tissues, our understanding of how these mutations correlate to carcinogenic progression is lacking. Tools to assess early mutations are needed to identify high cancer risk in normal-appearing tissues. Numerous studies have highlighted the efficacy of dietary prevention treatments in bladder cancer [47,48]. However, these methods are widely underutilized. Identifying patients at high risk for developing cancer allows time for the use of preventative interventions to potentially lessen the disease burden on these patients. The development of genomic tools to identify patients who will benefit most from preventative therapies will help usher in a new era of individualized targeted cancer prevention.

## 4. Materials and Methods

### 4.1. Datasets

All datasets used in this study were extracted from publicly available resources (Appendix A). The bladder cancer datasets were downloaded from cBioPortal (https://www.cbioportal.org/datasets accessed on 22 November 2022). A single non-muscle invasive bladder cancer dataset was available (MSK Eur Urol 2017) [21]. We chose the muscle-invasive bladder cancer dataset with the largest cohort of whole-exome sequenced samples (TCGA, Cell 2017) [22]. For the discovery of normal urothelium datasets, we searched the terms “bladder”, “mutation” and “normal” in PubMed. We screened 405 abstracts published within the last 5 years and identified the two largest datasets of normal urothelium [9,10]. The normal urothelium datasets were divided into two subsets: “high risk” and “low risk”. High-risk normal urothelium samples were collected from bladder cancer patients undergoing either cystectomy or nephroureterectomy to remove cancer specimens. Histologically normal areas of tissue were extracted via laser capture microdissection [9,10]. Low-risk normal urothelium samples were microbiopsies collected from post-mortem donors with no history of bladder cancer [9]. The datasets used contained both clinical patient characteristics and single nucleotide variant analysis by DNA-seq. A summary of the datasets, including sample size, sample type, and sequencing parameters, can be found in Appendix A.

### 4.2. Development of Hotspot Panels

We have developed an algorithm to design targeted sequencing panels based on genomic areas that harbor high-frequency mutations [20]. This algorithm, hotSPOT, uses SNV data from any sample type and discovers highly mutated areas specific to the inputted dataset. The hotSPOT tool then considers each highly mutated area and identifies an optimal combination of sequencing amplicons to cover these areas. For this study, we developed two distinct hotspot panels for MIBC and NMIBC (Appendix A). Both panels were created using our software tool hotSPOT. The hotSPOT tool is a published RShiny web application and may be accessed at https://rpccc-paraghlab-sgrant.shinyapps.io/hotspot/.

### 4.3. Analysis in R

All computational analyses were done in R version 4.1.1 [49]. Packages utilized for hotSPOT panel development include hash [50], rlist [51], and R.utils [52]. Packages for data visualization were ggplot2 [53], ggpubr [54], ggsignif [55], and ggrepel [56]. Packages for the development and testing of the cancer risk prediction model were caret [57], glmnet [58], neuralnet [59], randomForest [60], and pROC [61]. All statistical analyses were conducted using R version 4.1.1 ‘stats’ package [49]. A Pearson correlation was calculated to assess the correlation between hotspots. Statistical significance was calculated using a Wilcoxon signed-rank test.

## 5. Conclusions

The rapidly expanding field of cancer prevention offers a promising new expanding frontier for enhancing patient outcomes. Although numerous studies have reported data on mutation burdens in bladder cancer and normal urothelium, the correlation between mutations and carcinogenic progression remains poorly understood. Prior works comparing high- and low-risk normal urothelium have revealed a higher overall mutation frequency in high-risk samples; however, definitive markers of cancer risk remain elusive. In this study, our computational tool hotSPOT was employed to independently analyze NMIBC and MIBC datasets. Remarkably, all four hotspots identified in NMIBC were also present in MIBC, despite originating from distinct datasets, and we found the hotspots unique to MIBC as indicators of disease progression. The predictive ability of the MIBC hotspot burden in terms of progression is also supported by the observation that our most effective predictive model incorporated a combination of individual risk factors and mutations within MIBC hotspots. In summary, our findings illuminate novel genomic regions that may play a pivotal role in carcinogenic progression, demonstrating the potential of mutation hotspots to contribute to the development of risk assessment tools for healthy individuals. As our understanding of these genomic landscapes deepens, more effective strategies for cancer prevention and early detection may emerge.

## Figures and Tables

**Figure 1 ijms-24-07852-f001:**
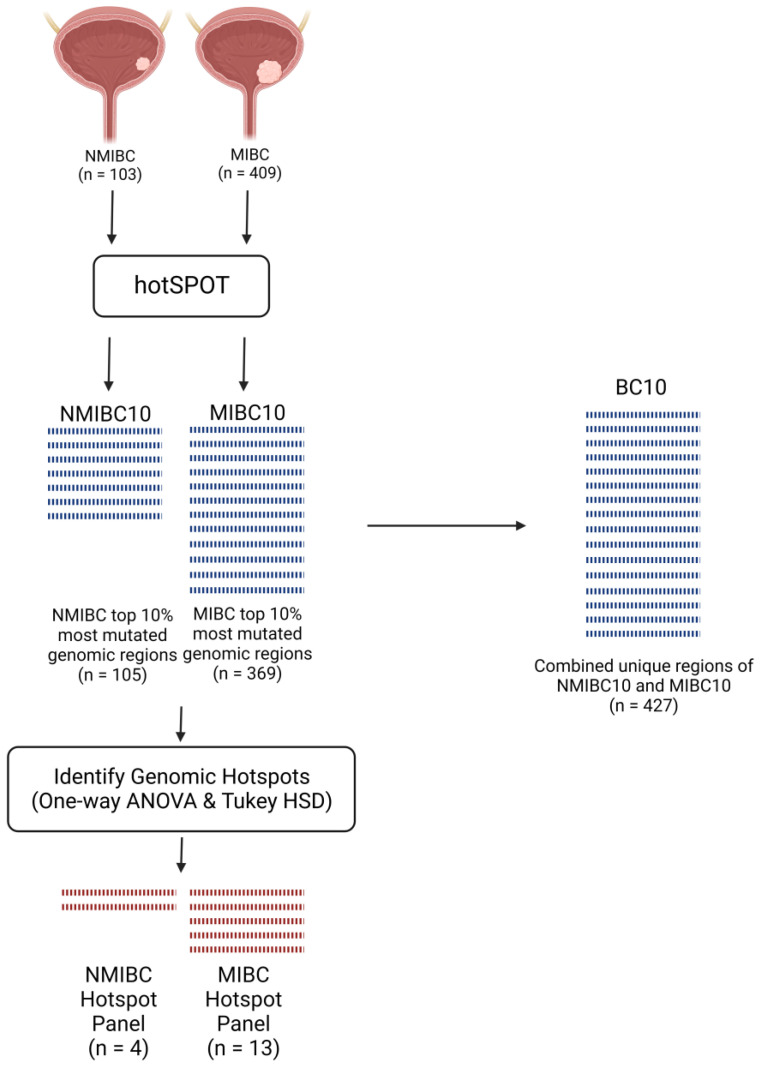
Schematic overview of the establishment of NMIBC and MIBC hotspot panels.

**Figure 2 ijms-24-07852-f002:**
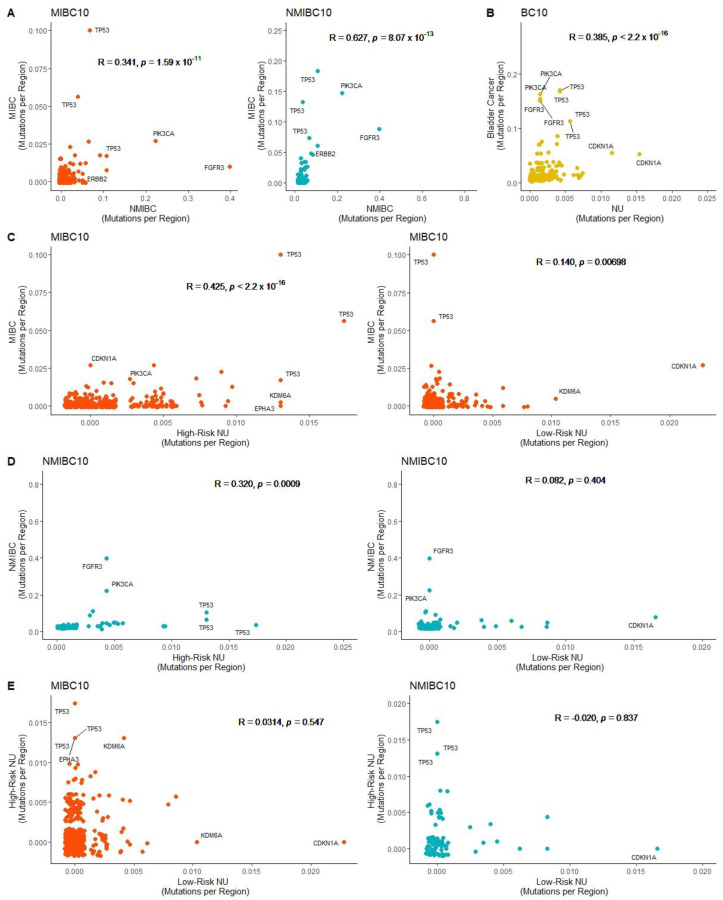
Comparison of mutation burden within bladder cancer hotspots (**A**) Pearson correlation between mutation frequency of NMIBC and MIBC within NMIBC10 and MIBC10 panels. (**B**) Pearson correlation between mutation frequency of NMIBC and MIBC combined and normal urothelium (NU) within BC10 regions. (**C**) Pearson correlation between mutation frequency of MIBC and low-risk NU, MIBC, and high-risk NU within MIBC10 (**D**) Pearson correlation between mutation frequency of NMIBC and low-risk NU, NMIBC, and high-risk NU within NMIBC10. (**E**) Pearson correlation between mutation frequency of low-risk NU and high-risk NU within MIBC10 and NMIBC10, respectively.

**Figure 3 ijms-24-07852-f003:**
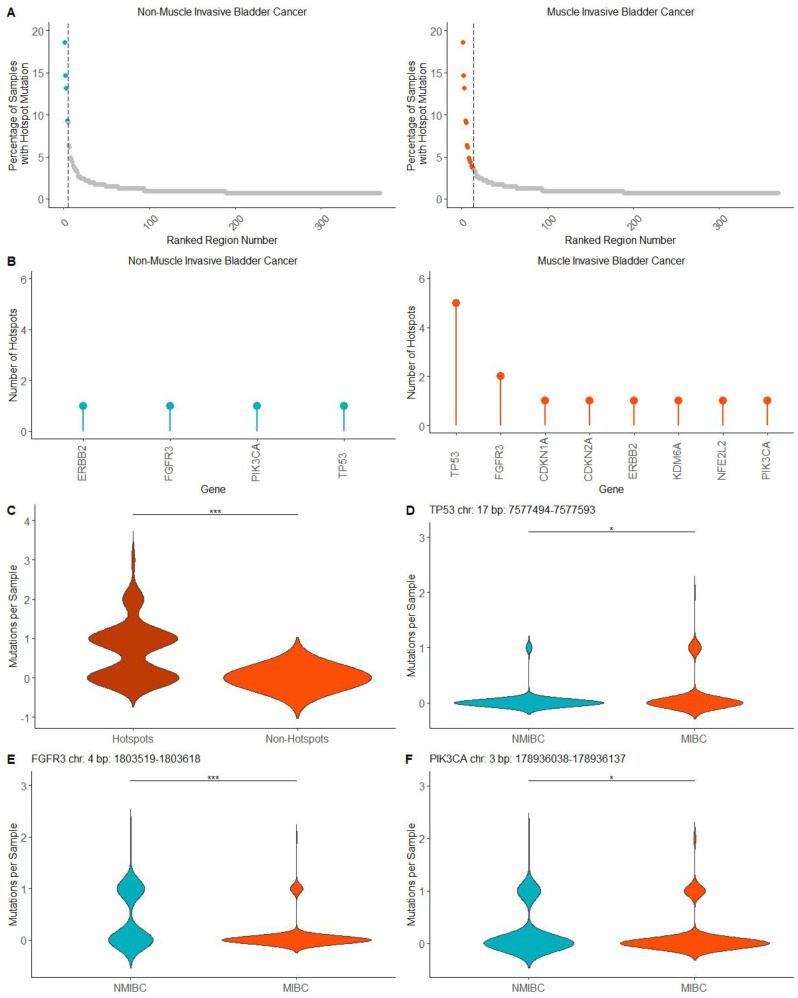
Hotspots in NMIBC and MIBC. (**A**) Percentage of samples with a hotspot mutation for NMIBC and MIBC hotspot panels compared with all genomic regions included in NMIBC10 and MIBC10. (**B**) Frequency of hotspots per gene in NMIBC and MIBC hotspot panels (**C**) Comparison of mutation capture between MIBC hotspots and an equal-sized genomic panel of minimally MIBC mutated regions in external validation MIBC dataset. (**D**) *TP53* hotspot in MIBC hotspot panel with significantly higher mutations in MIBC samples compared with NMIBC samples. (**E**) *FGFR3* hotspot shared between MIBC and NMIBC hotspot panels with significantly higher mutations in NMIBC samples compared with MIBC samples. (**F**) *PIK3CA* hotspot shared between MIBC and NMIBC hotspot panels with significantly higher mutations in NMIBC samples compared with MIBC samples. Wilcoxon signed-rank test (*: *p* < 0.05, ***: *p* < 0.001).

**Figure 4 ijms-24-07852-f004:**
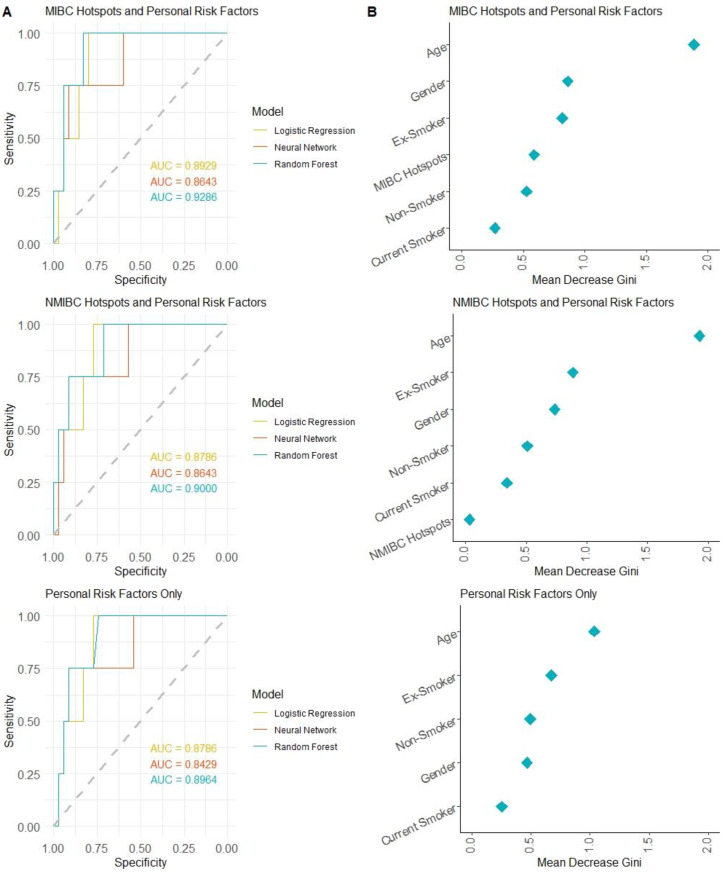
Hotspots in bladder urothelial carcinoma as predictive markers of carcinogenic risk. (**A**) ROC curve of classification model prediction accuracy based on the test dataset for logistic regression, neural network, and random forest models. (**B**) Variable importance plot of the hotspot and clinical variables for cancer risk prediction in the random forest classification model.

## Data Availability

Access to the hotSPOT web application is available at https://rpccc-paraghlab-sgrant.shinyapps.io/hotspot/ (accessed on 7 March 2023). Access to computational analysis from this study is available at: https://github.com/sydney-grant/Urothelium-Hotspots (accessed on 7 March 2023).

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
