# Peer review of "Mutation Hotspots Found in Bladder Cancer Aid Prediction of Carcinogenic Risk in Normal Urothelium"

_ijms, 2023, doi:10.3390/ijms24097852_

Round 1
Reviewer 1 Report
This is an interesting study that evaluated genetic mutations in normal urothelium. However, two concerns were raised.
1. Smoking is an environmental risk factor for bladder cancer. The authors should discuss the association of smoking with the genomic alterations detected in normal urothelium.
2. The difference between Western patients and non-Western patients is of interest of readers. Are there ethnic differences? The authors are encouraged to add more discussion in details on this point.
Author Response
Reviewer #1
Thank you for reviewing our manuscript and for your helpful critiques. We made the recommended modifications.
Comments and Suggestions for Authors
- Smoking is an environmental risk factor for bladder cancer. The authors should discuss the association of smoking with the genomic alterations detected in normal urothelium.
Additional text has been added to the discussion regarding smoking as a risk factor for bladder cancer, its implications on genomic alterations and relevancy to our study (lines 325-335).
- The difference between Western patients and non-Western patients is of interest of readers. Are there ethnic differences? The authors are encouraged to add more discussion in details on this point.
Additional text has been added to the discussion regarding the differences in populations between the normal urothelium datasets used for this study (lines 348-355).
Reviewer 2 Report
The paper investigated the aid of mutation hotspots found in bladder cancer to predict carcinogenic risk in normal urothelium.
The paper is clearly written and I found it very interesting to read.
Additional comments:
1) Figures 3B, D and E are not mentioned in the text.
2) According to the Figure 3E, FGF3 has higher mutations per sample in MIBC than NMIBC, but the figure legend and text states the opposite.
Author Response
Reviewer #2:
Thank you very much for reviewing our manuscript. We have made corrections to the manuscript based on your comments.
Comments and Suggestions for Authors
- Figures 3B, D and E are not mentioned in the text.
All labels relating to Figure 3 are now included in the results text (lines 176-190).
- According to the Figure 3E, FGF3 has higher mutations per sample in MIBC than NMIBC, but the figure legend and text states the opposite.
Thank you for pointing out this mistake in Figure 3E. The data included in this graph was incorrect, and this graph has been revised to include the correct data, showing that FGFR3 has higher mutations per sample in NMIBC (lines 208-210).
Author Response
Reviewer #3:
Thank you for reviewing our manuscript and for all your recommendations aimed at improving our manuscript. Please see the details below.
Comments and Suggestions for Authors
- Abstract of the study does not provide the novelty of this study, which is unique in the
study that has not been done and is observed for the first time
The text of the abstract has been updated to include a statement regarding the novelty of our study (lines 23-28).
- Sections and sub-sections of the study in the manuscript are not organized fairly.
Thank you for your suggestion, we have structured our manuscript to meet the structure guidelines for IJMS. A supplementary figure explaining the workflow of this study and the order of figures has been added.
- I recommend the authors discuss the methodology of the manuscript in the form of workflow. Supplementary Figure 1. Bladder cancer and normal urothelium dataset selection can be included in the main manuscript.
A supplementary figure regarding the workflow of the study has been added (lines 372-373).
- Authors may discuss more about the software tool “hotspot” developed by them; its screenshot can be given showing a test case. Are there any there any other similar tools that are already available? In addition, the link shared for a tool is not opening. May be authors can provide, how to use this tool in supplementary.
A statement regarding our previous publication which describes the methodology and usage of hotSPOT has been added to the text (line 94).
- I suggest authors provide conclusions to highlight the major findings and novelty of the study, separately from the discussion.
A conclusion highlighting the main findings and novelty of the study has been added (lines 410-426).
- Authors have applied a machine learning approach and used three algorithms namely logistic regression, neural network, and random forest. Can authors describe why only these three particular algorithms were selected for study?
An explanation of the selection of machine learning models has been added to the text (lines 228-234).
Round 2
Reviewer 1 Report
The authors have responded appropriately to my concerns, providing additional data.
This reviewer thinks this paper is improved.